# Finite-Time Analysis of Projected Langevin Monte Carlo

**Sébastien Bubeck**
Microsoft Research
sebubeck@microsoft.com

**Ronen Eldan**
Weizmann Institute
roneneldan@gmail.com

**Joseph Lehec**
Université Paris-Dauphine
lehec@ceremade.dauphine.fr

## Abstract

We analyze the projected Langevin Monte Carlo (LMC) algorithm, a close cousin of projected Stochastic Gradient Descent (SGD). We show that LMC allows to sample in polynomial time from a posterior distribution restricted to a convex body and with concave log-likelihood. This gives the first Markov chain to sample from a log-concave distribution with a first-order oracle, as the existing chains with provable guarantees (lattice walk, ball walk and hit-and-run) require a zeroth-order oracle. Our proof uses elementary concepts from stochastic calculus which could be useful more generally to understand SGD and its variants.

## 1 Introduction

A fundamental primitive in Bayesian learning is the ability to sample from the posterior distribution. Similarly to the situation in optimization, convexity is a key property to obtain algorithms with provable guarantees for this task. Indeed several Markov Chain Monte Carlo methods have been analyzed for the case where the posterior distribution is supported on a convex set, and the negative log-likelihood is convex. This is usually referred to as the problem of sampling from a log-concave distribution. In this paper we propose and analyze a new Markov chain for this problem which could have several advantages over existing chains for machine learning applications. We describe formally our contribution in Section 1.1. Then in Section 1.2 we explain how this contribution relates to various line of work in different fields such as theoretical computer science, statistics, stochastic approximation, and machine learning.

### 1.1 Main result

Let $K \subset \mathbb{R}^n$ be a convex set such that $0 \in K$, $K$ contains a Euclidean ball of radius $r > 0$ and is contained in a Euclidean ball of radius $R$. Denote $\mathcal{P}_K$ the Euclidean projection on $K$ (i.e., $\mathcal{P}_K(x) = \operatorname{argmin}_{y \in K} |x - y|$ where $|\cdot|$ denotes the Euclidean norm in $\mathbb{R}^n$), and $\|\cdot\|_K$ the gauge of $K$ defined by

$$\|x\|_K = \inf\{t \geq 0; \; x \in tK\}, \quad x \in \mathbb{R}^n.$$

Let $f : K \to \mathbb{R}$ be a $L$-Lipschitz and $\beta$-smooth convex function, that is $f$ is differentiable and satisfies $\forall x, y \in K, |\nabla f(x) - \nabla f(y)| \leq \beta |x - y|$, and $|\nabla f(x)| \leq L$. We are interested in the problem of sampling from the probability measure $\mu$ on $\mathbb{R}^n$ whose density with respect to the Lebesgue measure is given by:

$$\frac{d\mu}{dx} = \frac{1}{Z} \exp(-f(x)) \mathbb{1}\{x \in K\}, \quad \text{where} \quad Z = \int_{y \in K} \exp(-f(y)) dy.$$

We denote $m = \mathbb{E}_\mu |X|$, and $M = \mathbb{E}\left[\|\theta\|_K\right]$, where $\theta$ is uniform on the sphere $\mathbb{S}^{n-1} = \{x \in \mathbb{R}^n : |x| = 1\}$.

In this paper we study the following Markov chain, which depends on a parameter $\eta > 0$, and where $\xi_1, \xi_2, \ldots$ is an i.i.d. sequence of standard Gaussian random variables in $\mathbb{R}^n$, and $\overline{X}_0 = 0$,

$$\overline{X}_{k+1} = \mathcal{P}_K\left(\overline{X}_k - \frac{\eta}{2}\nabla f(\overline{X}_k) + \sqrt{\eta}\xi_k\right). \tag{1}$$

We call the chain (1) *projected Langevin Monte Carlo* (LMC).

Recall that the total variation distance between two measures $\mu, \nu$ is defined as $\mathrm{TV}(\mu, \nu) = \sup_A |\mu(A) - \nu(A)|$ where the supremum is over all measurable sets $A$. With a slight abuse of notation we sometimes write $\mathrm{TV}(X, \nu)$ where $X$ is a random variable distributed according to $\mu$. The notation $v_n = \widetilde{O}(u_n)$ (respectively $\widetilde{\Omega}$) means that there exists $c \in \mathbb{R}, C > 0$ such that $v_n \leq Cu_n \log^c(u_n)$ (respectively $\geq$).

Our main result shows that for an appropriately chosen step-size and number of iterations, one has convergence in total variation distance of the iterates $(\overline{X}_k)$ to the target distribution $\mu$.

**Theorem 1** *Let $\varepsilon > 0$. One has $\mathrm{TV}(\overline{X}_N, \mu) \leq \varepsilon$ provided that $\eta = \frac{1}{N}\left(\frac{m}{\varepsilon}\right)^2$ and*

$$N = \widetilde{\Omega}\left((n + RL)^2(M + L/r)^2 nm^6 \max\left(\frac{1}{\varepsilon^{16}}\left(\frac{n + RL}{r}\right)^6, \frac{1}{\varepsilon^{22}}\left(\beta m(L + \sqrt{R})\right)^8\right)\right).$$

Note that by viewing $\beta, L, r$ as numerical constants, using $M \leq 1/r$, and assuming $R \leq n$ and $m \leq n^{3/4}$, the bound reads

$$N = \widetilde{\Omega}\left(\frac{n^9 m^6}{\varepsilon^{22}}\right).$$

Observe also that if $f$ is constant, that is $\mu$ is the uniform measure on $K$, then $L = 0$, $m \leq \sqrt{n}$, and one can show that $M = \widetilde{O}(1/\sqrt{n})$, which yields the bound:

$$N = \widetilde{\Omega}\left(\left(\frac{n}{\varepsilon^2}\right)^{11}\right).$$

## 1.2 Context and related works

There is a long line of works in theoretical computer science proving results similar to Theorem 1, starting with the breakthough result of Dyer et al. [1991] who showed that the lattice walk mixes in $\widetilde{O}(n^{23})$ steps. The current record for the mixing time is obtained by Lovász and Vempala [2007], who show a bound of $\widetilde{O}(n^4)$ for the hit-and-run walk. These chains (as well as other popular chains such as the ball walk or the Dikin walk, see e.g. Kannan and Narayanan [2012] and references therein) all require a *zeroth-order oracle* for the potential $f$, that is given $x$ one can calculate the value $f(x)$. On the other hand our proposed chain (1) works with a *first-order oracle*, that is given $x$ one can calculate the value of $\nabla f(x)$. The difference between zeroth-order oracle and first-order oracle has been extensively studied in the optimization literature (e.g., Nemirovski and Yudin [1983]), but it has been largely ignored in the literature on polynomial-time sampling algorithms. We also note that hit-and-run and LMC are the only chains which are rapidly mixing from any starting point (see Lovász and Vempala [2006]), though they have this property for seemingly very different reasons. When initialized in a corner of the convex body, hit-and-run might take a long time to take a step, but once it moves it escapes very far (while a chain such as the ball walk would only do a small step). On the other hand LMC keeps moving at every step, even when initialized in a corner, thanks for the projection part of (1).

Our main motivation to study the chain (1) stems from its connection with the ubiquitous *stochastic gradient descent* (SGD) algorithm. In general this algorithm takes the form $x_{k+1} = \mathcal{P}_K(x_k - \eta\nabla f(x_k) + \varepsilon_k)$ where $\varepsilon_1, \varepsilon_2, \ldots$ is a centered i.i.d. sequence. Standard results in approximation theory, such as Robbins and Monro [1951], show that if the variance of the noise $\mathrm{Var}(\varepsilon_1)$ is of smaller order than the step-size $\eta$ then the iterates $(x_k)$ converge to the minimum of $f$ on $K$ (for a step-size decreasing sufficiently fast as a function of the number of iterations). For the specific noise

sequence that we study in (1), the variance is exactly equal to the step-size, which is why the chain deviates from its standard and well-understood behavior. We also note that other regimes where SGD does not converge to the minimum of $f$ have been studied in the optimization literature, such as the constant step-size case investigated in Pflug [1986], Bach and Moulines [2013].

The chain (1) is also closely related to a line of works in Bayesian statistics on Langevin Monte Carlo algorithms, starting essentially with Tweedie and Roberts [1996]. The focus there is on the unconstrained case, that is $K = \mathbb{R}^n$. In this simpler situation, a variant of Theorem 1 was proven in the recent paper Dalalyan [2014]. The latter result is the starting point of our work. A straightforward way to extend the analysis of Dalalyan to the constrained case is to run the unconstrained chain with an additional potential that diverges quickly as the distance from $x$ to $K$ increases. However it seems much more natural to study directly the chain (1). Unfortunately the techniques used in Dalalyan [2014] cannot deal with the singularities in the diffusion process which are introduced by the projection. As we explain in Section 1.3 our main contribution is to develop the appropriate machinery to study (1).

In the machine learning literature it was recently observed that Langevin Monte Carlo algorithms are particularly well-suited for large-scale applications because of the close connection to SGD. For instance Welling and Teh [2011] suggest to use mini-batch to compute approximate gradients instead of exact gradients in (1), and they call the resulting algorithm SGLD (Stochastic Gradient Langevin Dynamics). It is conceivable that the techniques developed in this paper could be used to analyze SGLD and its refinements introduced in Ahn et al. [2012]. We leave this as an open problem for future work. Another interesting direction for future work is to improve the polynomial dependency on the dimension and the inverse accuracy in Theorem 1 (our main goal here was to provide the simplest polynomial-time analysis).

## 1.3  Contribution and paper organization

As we pointed out above, Dalalyan [2014] proves the equivalent of Theorem 1 in the unconstrained case. His elegant approach is based on viewing LMC as a discretization of the diffusion process $dX_t = dW_t - \frac{1}{2}\nabla f(X_t)$, where $(W_t)$ is a Brownian motion. The analysis then proceeds in two steps, by deriving first the mixing time of the diffusion process, and then showing that the discretized process is 'close' to its continuous version. In Dalalyan [2014] the first step is particularly transparent as he assumes $\alpha$-strong convexity for the potential $f$, which in turns directly gives a mixing time of order $1/\alpha$. The second step is also simple once one realizes that LMC (without projection) can be viewed as the diffusion process $d\overline{X}_t = dW_t - \frac{1}{2}\nabla f(X_{\eta\lfloor\frac{t}{\eta}\rfloor})$. Using Pinsker's inequality and Girsanov's formula it is then a short calculation to show that the total variation distance between $\overline{X}_t$ and $X_t$ is small.

The constrained case presents several challenges, arising from the *reflection* of the diffusion process on the boundary of $K$, and from the lack of curvature in the potential (indeed the constant potential case is particularly important for us as it corresponds to $\mu$ being the uniform distribution on $K$). Rather than a simple Brownian motion with drift, LMC with projection can be viewed as the discretization of *reflected Brownian motion with drift*, which is a process of the form $dX_t = dW_t - \frac{1}{2}\nabla f(X_t)dt - \nu_t L(dt)$, where $X_t \in K, \forall t \geq 0$, $L$ is a measure supported on $\{t \geq 0 : X_t \in \partial K\}$, and $\nu_t$ is an outer normal unit vector of $K$ at $X_t$. The term $\nu_t L(dt)$ is referred to as the *Tanaka drift*. Following Dalalyan [2014] the analysis is again decomposed in two steps. We study the mixing time of the continuous process via a simple coupling argument, which crucially uses the convexity of $K$ and of the potential $f$. The main difficulty is in showing that the discretized process $(\overline{X}_t)$ is close to the continuous version $(X_t)$, as the Tanaka drift prevents us from a straightforward application of Girsanov's formula. Our approach around this issue is to first use a geometric argument to prove that the two processes are close in Wasserstein distance, and then to show that in fact for a reflected Brownian motion with drift one can deduce a total variation bound from a Wasserstein bound.

In this extended abstract we focus on the special case where $f$ is a constant function, that is $\mu$ is uniform on the convex body $K$. The generalization to an arbitrary smooth potential can be found in the supplementary material. The rest of the paper is organized as follows. Section 2 contains the main tehcnical arguments. We first remind the reader of Tanaka's construction (Tanaka [1979]) of reflected Brownian motion in Section 2.1. We present our geometric argument to bound the

Wasserstein distance between $(X_t)$ and $(\overline{X}_t)$ in Section 2.2, and we use our coupling argument to bound the mixing time of $(X_t)$ in Section 2.3. The derivation of a total variation bound from the Wasserstein bound is discussed in Section 2.4. Finally we conclude the paper in Section 3 with some preliminary experimental comparison between LMC and hit-and-run.

## 2 The constant potential case

In this section we derive the main arguments to prove Theorem 1 when $f$ is a constant function, that is $\nabla f = 0$. For a point $x \in \partial K$ we say that $\nu$ is an outer unit normal vector at $x$ if $|\nu| = 1$ and
$$\langle x - x', \nu \rangle \geq 0, \quad \forall x' \in K.$$
For $x \notin \partial K$ we say that $0$ is an outer unit normal at $x$. We define the support function $h_K$ of $K$ by
$$h_K(y) = \sup\{\langle x, y \rangle; \; x \in K\}, \quad y \in \mathbb{R}^n.$$
Note that $h_K$ is also the gauge function of the polar body of $K$.

### 2.1 The Skorokhod problem

Let $T \in \mathbb{R}_+ \cup \{+\infty\}$ and $w \colon [0,T) \to \mathbb{R}^n$ be a piecewise continuous path with $w(0) \in K$. We say that $x \colon [0,T) \to \mathbb{R}^n$ and $\varphi \colon [0,T) \to \mathbb{R}^n$ solve the Skorokhod problem for $w$ if one has $x(t) \in K, \forall t \in [0,T)$,
$$x(t) = w(t) + \varphi(t), \quad \forall t \in [0,T),$$
and furthermore $\varphi$ is of the form
$$\varphi(t) = -\int_0^t \nu_s \, L(ds), \quad \forall t \in [0,T),$$
where $\nu_s$ is an outer unit normal at $x(s)$, and $L$ is a measure on $[0,T]$ supported on the set $\{t \in [0,T) : \; x(t) \in \partial K\}$.

The path $x$ is called the *reflection* of $w$ at the boundary of $K$, and the measure $L$ is called the *local time* of $x$ at the boundary of $K$. Skorokhod showed the existence of such a a pair $(x, \varphi)$ in dimension 1 in Skorokhod [1961], and Tanaka extended this result to convex sets in higher dimensions in Tanaka [1979]. Furthermore Tanaka also showed that the solution is unique, and if $w$ is continuous then so is $x$ and $\varphi$. In particular the reflected Brownian motion in $K$, denoted $(X_t)$, is defined as the reflection of the standard Brownian motion $(W_t)$ at the boundary of $K$ (existence follows by continuity of $W_t$). Observe that by Itô's formula, for any smooth function $g$ on $\mathbb{R}^n$,
$$g(X_t) - g(X_0) = \int_0^t \langle \nabla g(X_s), dW_s \rangle + \frac{1}{2} \int_0^t \Delta g(X_s) \, ds - \int_0^t \langle \nabla g(X_s), \nu_s \rangle \, L(ds). \quad (2)$$

To get a sense of what a solution typically looks like, let us work out the case where $w$ is piecewise constant (this will also be useful to realize that LMC can be viewed as the solution to a Skorokhod problem). For a sequence $g_1 \ldots g_N \in \mathbb{R}^n$, and for $\eta > 0$, we consider the path:
$$w(t) = \sum_{k=1}^N g_k \, \mathbb{1}\{t \geq k\eta\}, \qquad t \in [0, (N+1)\eta).$$
Define $(x_k)_{k=0,\ldots,N}$ inductively by $x_0 = 0$ and
$$x_{k+1} = \mathcal{P}_K(x_k + g_k).$$
It is easy to verify that the solution to the Skorokhod problem for $w$ is given by $x(t) = x_{\eta \lfloor \frac{t}{\eta} \rfloor}$ and $\varphi(t) = -\int_0^t \nu_s \, L(ds)$, where the measure $L$ is defined by (denoting $\delta_s$ for a dirac at $s$)
$$L = \sum_{k=1}^N |x_k + g_k - \mathcal{P}_K(x_k + g_k)| \delta_{k\eta},$$
and for $s = k\eta$,
$$\nu_s = \frac{x_k + g_k - \mathcal{P}_K(x_k + g_k)}{|x_k + g_k - \mathcal{P}_K(x_k + g_k)|}.$$

## 2.2 Discretization of reflected Brownian motion

Given the discussion above, it is clear that when $f$ is a constant function, the chain (1) can be viewed as the reflection $(\overline{X}_t)$ of a discretized Brownian motion $\overline{W}_t := W_{\eta \lfloor \frac{t}{\eta} \rfloor}$ at the boundary of $K$ (more precisely the value of $\overline{X}_{k\eta}$ coincides with the value of $\overline{X}_k$ as defined by (1)). It is rather clear that the discretized Brownian motion $(\overline{W}_t)$ is "close" to the path $(W_t)$, and we would like to carry this to the reflected paths $(\overline{X}_t)$ and $(X_t)$. The following lemma extracted from Tanaka [1979] allows to do exactly that.

**Lemma 1** *Let $w$ and $\overline{w}$ be piecewise continuous path and assume that $(x, \varphi)$ and $(\overline{x}, \overline{\varphi})$ solve the Skorokhod problems for $w$ and $\overline{w}$, respectively. Then for all time $t$ we have*

$$|x(t) - \overline{x}(t)|^2 \leq |w(t) - \overline{w}(t)|^2$$
$$+ 2 \int_0^t \langle w(t) - \overline{w}(t) - w(s) + \overline{w}(s), \varphi(ds) - \overline{\varphi}(ds) \rangle.$$

Applying the above lemma to the processes $(W_t)$ and $(\overline{W}_t)$ at time $T = N\eta$ yields (note that $W_T = \overline{W}_T$)

$$|X_T - \overline{X}_T|^2 \leq -2 \int_0^T \langle W_t - \overline{W}_t, \nu_t \rangle L(dt) + 2 \int_0^T \langle W_t - \overline{W}_t, \overline{\nu}_t \rangle \overline{L}(dt)$$

We claim that the second integral is equal to 0. Indeed, since the discretized process is constant on the intervals $[k\eta, (k+1)\eta)$ the local time $\overline{L}$ is a positive combination of Dirac point masses at

$$\eta, 2\eta, \ldots, N\eta.$$

On the other hand $W_{k\eta} = \overline{W}_{k\eta}$ for all integer $k$, hence the claim. Therefore

$$|X_T - \overline{X}_T|^2 \leq -2 \int_0^T \langle W_t - \overline{W}_t, \nu_t \rangle L(dt)$$

Using the inequality $\langle x, y \rangle \leq \|x\|_K h_K(y)$ we get

$$|X_T - \overline{X}_T|^2 \leq 2 \sup_{[0,T]} \|W_t - \overline{W}_T\|_K \int_0^T h_K(\nu_t) L(dt).$$

Taking the square root, expectation and using Cauchy–Schwarz we get

$$\mathbb{E}\left[|X_T - \overline{X}_T|\right]^2 \leq 2 \mathbb{E}\left[\sup_{[0,T]} \|W_t - \overline{W}_T\|_K\right] \mathbb{E}\left[\int_0^T h_K(\nu_t) L(dt)\right]. \tag{3}$$

The next two lemmas deal with each term in the right hand side of the above equation, and they will show that there exists a universal constant $C$ such that

$$\mathbb{E}\left[|X_T - \overline{X}_T|\right] \leq C \left(\eta \log(T/\eta)\right)^{1/4} n^{3/4} T^{1/2} M^{1/2}. \tag{4}$$

We discuss why the above bound implies a total variation bound in Section 2.4.

**Lemma 2** *We have, for all $t > 0$*

$$\mathbb{E}\left[\int_0^t h_K(\nu_s) L(ds)\right] \leq \frac{nt}{2}.$$

**Proof** By Itô's formula

$$d|X_t|^2 = 2\langle X_t, dW_t \rangle + n\,dt - 2\langle X_t, \nu_t \rangle L(dt).$$

Now observe that by definition of the reflection, if $t$ is in the support of $L$ then

$$\langle X_t, \nu_t \rangle \geq \langle x, \nu_t \rangle, \quad \forall x \in K.$$

In other words $\langle X_t, \nu_t \rangle \geq h_K(\nu_t)$. Therefore

$$2 \int_0^t h_K(\nu_s) L(ds) \leq 2 \int_0^t \langle X_s, dW_s \rangle + nt + |X_0|^2 - |X_t|^2.$$

The first term of the right–hand side is a martingale, so using that $X_0 = 0$ and taking expectation we get the result. ∎

**Lemma 3** *There exists a universal constant $C$ such that*

$$\mathbb{E}\left[\sup_{[0,T]} \|W_t - \overline{W}_t\|_K\right] \leq C\, M\, \sqrt{n\eta \log(T/\eta)}.$$

**Proof** Note that

$$\mathbb{E}\left[\sup_{[0,T]} \|W_t - \overline{W}_t\|_K\right] = \mathbb{E}\left[\max_{0 \leq i \leq N-1} Y_i\right]$$

where

$$Y_i = \sup_{t \in [i\eta, (i+1)\eta)} \|W_t - W_{i\eta}\|_K.$$

Observe that the variables $(Y_i)$ are identically distributed, let $p \geq 1$ and write

$$\mathbb{E}\left[\max_{i \leq N-1} Y_i\right] \leq \mathbb{E}\left[\left(\sum_{i=0}^{N-1} |Y_i|^p\right)^{1/p}\right] \leq N^{1/p} \|Y_0\|_p.$$

We claim that

$$\|Y_0\|_p \leq C\sqrt{p\, n\, \eta}\, M \tag{5}$$

for some constant $C$, and for all $p \geq 2$. Taking this for granted and choosing $p = \log(N)$ in the previous inequality yields the result (recall that $N = T/\eta$). So it is enough to prove (5). Observe that since $(W_t)$ is a martingale, the process $M_t = \|W_t\|_K$ is a sub–martingale. By Doob's maximal inequality

$$\|Y_0\|_p = \|\sup_{[0,\eta]} M_t\|_p \leq 2\|M_\eta\|_p,$$

for every $p \geq 2$. Letting $\gamma_n$ be the standard Gaussian measure on $\mathbb{R}^n$ and using Khintchin's inequality we get

$$\|M_\eta\|_p = \sqrt{\eta}\left(\int_{\mathbb{R}^n} \|x\|_K^p\, \gamma_n(dx)\right)^{1/p} \leq C\sqrt{p\eta} \int_{\mathbb{R}^n} \|x\|_K\, \gamma_n(dx).$$

Lastly, integrating in polar coordinate, it is easily seen that

$$\int_{\mathbb{R}^n} \|x\|_K\, \gamma_n(dx) \leq C\sqrt{n}\, M.$$

$\blacksquare$

## 2.3 A mixing time estimate for the reflected Brownian motion

Given a probability measure $\nu$ supported on $K$, we let $\nu P_t$ be the law of $X_t$ when $X_0$ has law $\nu$. The following lemma is the key result to estimate the mixing time of the process $(X_t)$.

**Lemma 4** *Let $x, x' \in K$*

$$\mathrm{TV}(\delta_x P_t, \delta_{x'} P_t) \leq \frac{|x - x'|}{\sqrt{2\pi t}}.$$

The above result clearly implies that for a probability measure $\nu$ on $K$, $\mathrm{TV}(\delta_0 P_t, \nu P_t) \leq \frac{\int_K |x|\, \nu(dx)}{\sqrt{2\pi t}}$. Since $\mu$ (the uniform measure on $K$) is stationary for reflected Brownian motion, we obtain

$$\mathrm{TV}(\delta_0 P_t, \mu) \leq \frac{m}{\sqrt{2\pi t}}. \tag{6}$$

In other words, starting from 0, the mixing time of $(X_t)$ is of order $m^2$. We now turn to the proof of the above lemma.

**Proof** The proof is based on a coupling argument. Let $(W_t)$ be a Brownian motion starting from 0 and let $(X_t)$ be a reflected Brownian motion starting from $x$:

$$\begin{cases} X_0 = x \\ dX_t = dW_t - \nu_t\, L(dt) \end{cases}$$

where $(\nu_t)$ and $L$ satisfy the appropriate conditions. We construct a reflected Brownian motion $(X'_t)$ starting from $x'$ as follows. Let $\tau = \inf\{t \geq 0;\ X_t = X'_t\}$, and for $t < \tau$ let $S_t$ be the orthogonal reflection with respect to the hyperplane $(X_t - X'_t)^{\perp}$. Then up to time $\tau$, the process $(X'_t)$ is defined by

$$\begin{cases} X'_0 = x' \\ dX'_t = dW'_t - \nu'_t\, L'(dt) \\ dW'_t = S_t(dW_t) \end{cases}$$

where $L'$ is a measure supported on $\{t \leq \tau;\ X'_t \in \partial K\}$, and $\nu'_t$ is an outer unit normal at $X'_t$ for all such $t$. After time $\tau$ we just set $X'_t = X_t$. Since $S_t$ is an orthogonal map $(W'_t)$ is a Brownian motion and thus $(X'_t)$ is a reflected Brownian motion starting from $x'$. Therefore

$$\mathrm{TV}(\delta_x P_t, \delta_{x'} P_t) \leq \mathbb{P}(X_t \neq X'_t) = \mathbb{P}(\tau > t).$$

Observe that on $[0, \tau)$

$$dW_t - dW'_t = (\mathrm{I} - S_t)(dW_t) = 2\langle V_t, dW_t \rangle V_t,$$

where $V_t = \frac{X_t - X'_t}{|X_t - X'_t|}$. So

$$d(X_t - X'_t) = 2\langle V_t, dW_t \rangle V_t - \nu_t\, L(dt) + \nu'_t\, L'(dt) = 2(dB_t)\, V_t - \nu_t\, L(dt) + \nu'_t\, L'(dt),$$

where

$$B_t = \int_0^t \langle V_s, dW_s \rangle, \quad \text{on } [0, \tau).$$

Observe that $(B_t)$ is a one–dimensional Brownian motion. Itô's formula then gives

$$\begin{aligned} dg(X_t - X'_t) = {} & 2\langle \nabla g(X_t - X'_t), V_t \rangle\, dB_t - \langle \nabla g(X_t - X'_t), \nu_t \rangle\, L(dt) \\ & + \langle \nabla g(X_t - X'_t), \nu't \rangle\, L'(dt) + 2\nabla^2 g(X_t - X'_t)(V_t, V_t)\, dt, \end{aligned}$$

for every smooth function $g$ on $\mathbb{R}^n$. Now if $g(x) = |x|$ then

$$\nabla g(X_t - X'_t) = V_t$$

so $\langle \nabla g(X_t - X'_t), V_t \rangle = 1$, $\langle \nabla g(X_t - X'_t), \nu_t \rangle \geq 0$ on the support of $L$, and $\langle \nabla g(X_t - X'_t), \nu'_t \rangle \leq 0$ on the support of $L'$. Moreover $\nabla^2 g(X_t - X'_t) = \frac{1}{|X_t - Y_t|} P_{(X_t - Y_t)^{\perp}}$ where $P_{x^{\perp}}$ denotes the orthogonal projection on $x^{\perp}$. In particular $\nabla^2 g(X_t - Y_t)(V_t) = 0$. We obtain $|X_t - X'_t| \leq |x - x'| + 2B_t$, on $[0, \tau)$. Therefore $\mathbb{P}(\tau > t) \leq \mathbb{P}(\tau' > t)$ where $\tau'$ is the first time the Brownian motion $(B_t)$ hits the value $-|x - x'|/2$. Now by the reflection principle

$$\mathbb{P}(\tau' > t) = 2\, \mathbb{P}\left(0 \leq 2\, B_t < |x - x'|\right) \leq \frac{|x - x'|}{\sqrt{2\pi t}}.$$

∎

## 2.4 From Wasserstein distance to total variation

To conclude it remains to derive a total variation bound between $X_T$ and $\overline{X}_T$ using (4). The details of this step are deferred to the supplementary material where we consider the case of a general log-concave distribution. The intuition goes as follows: the processes $(X_{T+s})_{s \geq 0}$ and $(\overline{X}_{T+s})_{s \geq 0}$ both evolve according to a Brownian motion until the first time $s$ that one process undergoes a reflection. But if $T$ is large enough and $\eta$ is small enough then one can easily get from (4) (and the fact that the uniform measure does not put too much mass close to the boundary) that $X_T$ and $\overline{X}_T$ are much closer to each other than they are to the boundary of $K$. This implies that one can couple them (just as in Section 2.3) so that they meet before one of them hits the boundary.

## 3 Experiments

Comparing different Markov Chain Monte Carlo algorithms is a challenging problem in and of itself. Here we choose the following simple comparison procedure based on the volume algorithm

developed in Cousins and Vempala [2014]. This algorithm, whose objective is to compute the volume of a given convex set $K$, procedes in phases. In each phase $\ell$ it estimates the mean of a certain function under a multivariate Gaussian restricted to $K$ with (unrestricted) covariance $\sigma_\ell \mathrm{I}_n$. Cousins and Vempala provide a Matlab implementation of the entire algorithm, where in each phase the target mean is estimated by sampling from the truncated Gaussian using the hit-and-run (H&R) chain. We implemented the same procedure with LMC instead of H&R, and we choose the step-size $\eta = 1/(\beta n^2)$, where $\beta$ is the smoothness parameter of the underlying log-concave distribution (in particular here $\beta = 1/\sigma_\ell^2$). The intuition for the choice of the step-size is as follows: the scaling in inverse smoothness comes from the optimization literature, while the scaling in inverse dimension squared comes from the analysis in the unconstrained case in Dalalyan [2014].

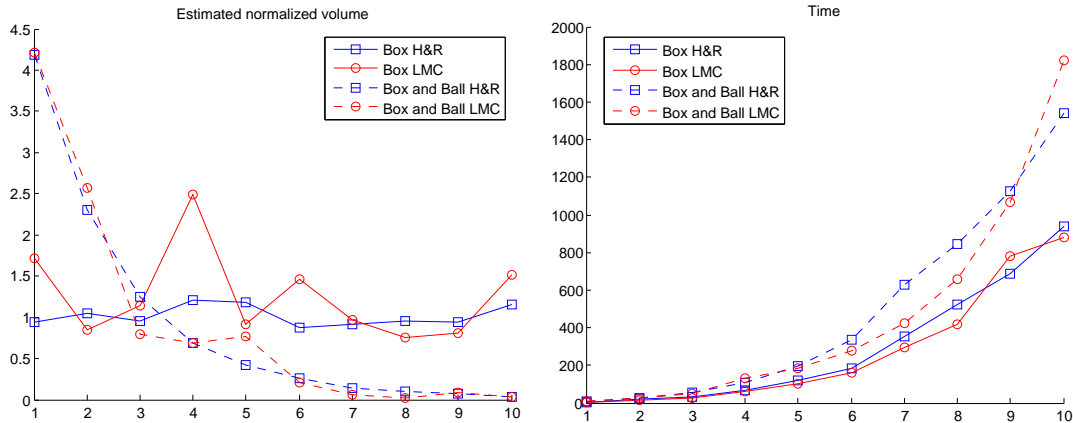

We ran the volume algorithm with both H&R and LMC on the following set of convex bodies: $K = [-1, 1]^n$ (referred to as the "Box") and $K = [-1, 1]^n \cap \left( \frac{\sqrt{n}}{2} \mathbb{B}^n \right)$ (referred to as the "Box and Ball"), where $n = 10 \times k, k = 1, \dots, 10$. The computed volume (normalized by $2^n$ for the "Box" and by $0.2 \times 2^n$ for the "Box and Ball") as well as the clock time (in seconds) to terminate are reported in the figure above. From these experiments it seems that LMC and H&R roughly compute similar values for the volume (with H&R being slightly more accurate), and LMC is almost always a bit faster. These results are encouraging, but much more extensive experiments are needed to decide if LMC is indeed a competitor to H&R in practice.

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
