[Reviews · NeurIPS 2015]

Submitted by Assigned_Reviewer_1

The analysis by the authors is highly technical and impressive.

The issues with the paper are: 1. There are many areas where facts are stated and a citation is expected, but none is provided. For example, "Indeed several Markov Chain Monte Carlo methods have been analyzed for the case where the posterior distribution is supported on a convex set, and the negative log-likelihood is convex." As well as, "... convexity is a key property to obtain algorithms with provable guarantees for this task."

2. It would be interesting if the results section compared to more standard MCMC algorithms on interesting inference problems rather than just "volume estimation" benchmarks. The performance gains should also be specified quantitatively rather than just graphically.

3. There is no conclusions section (even a few sentences). That is not a very polished way to end a paper.

4. There are no axis labels on the plots. Error bars would be nice too.

Other comments: 1. It would be nice if the authors could elaborate on what exactly they mean with "Unfortunately the techniques used in Dalalyan [2014] cannot deal with the singularities in the diffusion process ...". What singularities?

2. It appears there might be a typo on line 307. Is the C supposed to appear in the inequality? Would that make Lemma 3 contain C^2?
Summary: The authors do a heavy theoretical analysis to provide guarantees on the accuracy of Langevin Monte Carlo methods. Their main result shows a big-O like result relating the number of samples required, the dimension of the space, and the desired accuracy (as measured in total variation distance between the target distribution and the one actually sampled from).

Submitted by Assigned_Reviewer_2

The paper presents a finite-time analysis of new MC algorithm called Projected LMC.

The optimization counterpart of Projected LMC is stochastic gradient descent (SGD). The main theoretical result states that for an appropriately chosen step-size and after a large number of iterations, the distribution of the samples from Projected LMC is arbitrarily close to the target distribution log-concave density.

The main contribution here is to show that the step-size of Projected LMC is similar to the step-size in SGD and that the maximum number of iteration of Projected-LMC depend polynomially on the dimension of the space (up to a logarithmic factor).

Further, the proposed algorithm does not involve evaluation of the density, but its gradient.

This is extremely useful in cases where the normalization constant is not tractable.

The paper is clear and well-written. It introduces a new theoretical analysis of Projected LMC algorithm for sampling from distributions with log-concave densities that involves evaluation of the first order oracle.
Summary: This work introduces a finite-time analysis of the projected Langevin Monte Carlo (LMC) algorithm.

The approach first theoretical analysis of a MC algorithm to sample from a log-concave distribution that only requires values of the gradient of log-density.

Submitted by Assigned_Reviewer_3

The authors consider the projected Langevin Monte-Carlo (PLMC) algorithm for sampling from a log-concave density with a compact support. They establish a polynomial-time bound for sampling with a prescribed accuracy. The result is new, nontrivial and, in my opinion provides a very interesting contribution to the theory of computational statistics/machine learning.

I went through the proof (without reading the parts which are in the supplementary material) and checked all the details. I have found only a couple of small typos (cf. below). Not only the proof is correct, but also it contains a very elegant argument that makes use of the Skorohod embedding. I am sure any mathematically driven person will appreciate this argument.

Here are some minor remarks: - I recommend to add somewhere in the discussion that the PLMC needs not only a first-order oracle but also a projector to the compact set under consideration.

- Concerning the discussion on H&R in Section 1.2: the authors seem to claim that H&R requires a zeroth-order oracle, but my understanding is that H&M requires to sample from the restriction of the distribution on the segments. Could the authors comment on the relationship between these two tasks? - Beginning of section 2.1: why do you need to introduce the pair (x,phi)? It is my impression that it suffices to introduce x. Then you can remove the equation of line 184 and replace phi by x-w in the equation on line 187.

- line 210: "Dirac" instead of "dirac" -line 225: "paths" instead of "path" - line 244 and Eq (3): in these two formulae there is a \bar W_T that should be replaced by \bar W_t (small t instead of capital T). - line 298: please provide a precise reference for the version of the Khintchine inequality you use.

- line 321: could you provide a reference where it is clearly stated that the uniform distribution on a compact set is a stationary distribution for the reflected BM ?

N.B. In the confidence evaluation above I have put 4 (and not 5) just because I cannot be absolutely certain of my evaluation of the paper without carefully reading all the details of the proofs which are put in the supplementary material.

Summary: This papers contains a very nice theoretical result about the computational complexity of sampling from a compactly supported distribution with a log-concave density when a first-order oracle is available. This contribution is original and concerns an important topic which deserves to receive more attention.

Author Feedback
Author rebuttal: Reviewers 1, 5, and 6: Our work should be evaluated on its two main contributions. One is methodological: we show that Projected Stochastic Gradient Descent (which is ubiquitous in Machine Learning) can be used for the basic task of sampling from a posterior with a convex negative log-likelihood. This new point of view could have an impact in practice. Our second contribution is mathematical, as we introduce the use of stochastic calculus in the field of Machine Learning. This could have an impact for the analysis of other algorithms in Machine Learning.

Reviewer 3: We thank you for your encouraging comments! As you are the only reviewer with a confidence > 2, it would be very helpful if you could initiate a discussion with the other reviewers to convince them of the significance of our work.

Point by point answer to reviewer_1 comments:
1. We will add more citations in the introduction.
2. We completely agree with you, and as we state in the paper these are only preliminary experiments which should be viewed as a proof of concept. Our main contributions are methodological and conceptual (as we explained above).
3. We do not think that this particular paper needs a conclusion. We believe that Section 1 (and in particular Section 1.2) is rather clear in terms of what is left for future works.
4. You are right, we will add those.
Minor comments: 1. The projection step is in a sense a very "non-smooth" operation, which is a serious difficulty for the analysis of a stochastic differential equation. 2. This is not a typo, C is a universal constant whose value can change at each occurrence.